# Modeling neonatal immune response to *B. pertussis* identifies early B cell activation and differentiation

Soraya Matczak[1,�], Mirko Sadi[1,☉он], Pauline Leroux[1,‡], Pauline Labé[1,2,‡], Valérie Bouchez[1,3], Sandrine Schmutz[4], Valentina Libri[5], Valérie Seffer[5], Sophie Novault[4], Stevenn Volant[6], Sylvain Brisse[1,3], Milena Hasan[5], Darragh Duffy[7], Pierre Tonnerre[1,8,†], Julie Toubiana[☉1,2,3,†*]

**1** Institut Pasteur, Université Paris Cité, Biodiversity and Epidemiology of Bacterial Pathogens, Paris, France, **2** Department of General Pediatrics and Pediatric Infectious Diseases, Université Paris Cité, Hôpital Necker–Enfants malades, APHP, Paris, France, **3** Institut Pasteur, National Reference Center for Whooping Cough and other Bordetella infections, Paris, France, **4** Institut Pasteur, Université Paris Cité, Cytometry Platform, Paris, France, **5** Institut Pasteur, Université Paris Cité, Single Cell Biomarkers UTechS, Paris, France, **6** Institut Pasteur, Université Paris Cité, Bioinformatics and Biostatistics Hub, Paris, France, **7** Institut Pasteur, Université Paris Cité, Translational Immunology Unit, Paris, France, **8** Institut de Recherche Saint-Louis, Université Paris-Cité, Inserm U1342, Team ATIP-Avenir, Paris, France

☉ These authors contributed equally.
‡ These authors contributed equally.
† These authors share last authorship.
* julie.toubiana@pasteur.fr

## Abstract

Pertussis is typically more severe in neonates and young infants, with fulminant forms possibly linked to age-specific systemic immune responses. However, the early events that shape the development of the immune response in neonates are poorly understood, mainly due to limited sample access and the absence of human *ex vivo* infection models. Here, multi-omic profiling of an *ex vivo* whole-blood infection model was used to investigate systemic immune responses to *Bordetella pertussis*. We observed a stronger pro-inflammatory immune response in cord blood (CB) compared to adult blood (AB), marked by a hyperinflammatory cytokine signature and an early loss of myeloid-derived suppressor cells. B cell remodeling was observed in both blood types, characterized by an increased fraction of CD25 + B lymphocytes exhibiting an activated transitional phenotype, with upregulated expression of activation markers, chemokines and immunosuppressors. Transcriptomic analysis of whole blood revealed a skewed immune response favoring innate immune cell activation in CB. Our findings shed light on early immune responses to *B. pertussis*, paving the way for further exploration of immune pathways in the pathogenesis of this major public health threat.

**Data availability statement:** We generated all data set and provided extensive cytokine (S1 Dataset) and transcriptomic datasets in our Supporting information files (S2 Dataset). The protocol of the study and ethical approvals are available on the laboratory website (research.pasteur.fr/en/project/fr-pert-severe-definir-la-contribution-des-facteurs-microbiens-et-de-limmunite-de-lhote-a-la-gravite-de-la-co-queluche/).

**Funding:** JT received funding the Agence Nationale de Recherche (ANR-22-CE17-0014) and from Ville-de-Paris as part of the EMERGENCE program. PLe and SM received funding from the Fondation pour la Recherche Médicale (FRM), PLa and SM from Société de Pathologie Infectieuse de Langue Française (SPILF). The funders had no role in study design, data collection and analysis, decision to publish, or preparation of the manuscript.

**Competing interests:** The authors have declared that no competing interests exist.

## Author summary

Pertussis disease is mainly caused by the infection of the respiratory tract by the Gram-negative bacterium *Bordetella pertussis*. Despite global vaccination efforts, pertussis outbreaks continue to threaten neonates and young infants who face the most severe, potentially deadly, forms of the disease. The latter is characterized by unique clinical symptoms reflecting a specific immune response in infants. The neonatal immune system is naturally biased towards immuno-suppression to protect developing tissues, thus, however, increasing susceptibility to infections. To understand severe pertussis pathogenesis, our study characterized the immune responses in cord blood and adult blood to *ex vivo* stimulation with *B. pertussis*. We observed distinct immune response profiles; several immune-regulatory markers were differentially expressed, highlighting fundamental differences in host responses. Strikingly, a sepsis-like response, known to be associated with severe pertussis, was observed exclusively in cord blood. Our study sheds light on the poorly understood pathogenesis of severe pertussis disease in infants and identifies a specific immune profile in response to pertussis infection *ex vivo*. It therefore paves the way for future translational studies and represents a starting point for the exploration of new prevention and treatment strategies.

## Introduction

Whooping cough or pertussis is a highly contagious respiratory disease whose causative agent is *Bordetella pertussis* [1]. Despite acellular pertussis vaccine (aPV) implementation, a resurgence is being observed in many parts of the world, including in highly vaccinated populations [2]. Most pertussis morbidity is due to severe clinical forms in young infants and newborns that usually require admission in intensive care units (ICU) [3,4]. The most severe form, also called fulminant pertussis (FP), is a particular clinical phenotype of pertussis infection, exclusively observed in young infants and which associates acute respiratory distress syndrome, refractory pulmonary hypertension, encephalopathy, and/or hemodynamic instability, and elevated white blood cells [5,6]. The mechanisms underlying the clinical aspects of FP and associated high leukocytosis, are still poorly understood. One hypothesis is that a specific inflammatory and immune response of the infant or neonate might be involved, as observed for other pathogens [7].

Young infants show a diminished capacity to mount an effective immune response to infectious agents, including *B. pertussis*, resulting in enhanced susceptibility [8–12]. This impaired innate immunity could be partly explained by immunomodulation of CD71+ erythroid suppressor cells which are present in high numbers in neonates and capable of inhibiting innate responses to *B. pertussis* infection in neonatal mice by the expression of arginase II [13,14]. Other circulating regulatory cells that

persist from fetal life as regulatory T cells (Tregs), B cells (Bregs), and myeloid derived suppressor cells (MDSCs) remain elevated during the neonatal period. Neonatal cells also have a lower capacity to generate memory cells and Th1 effector responses [7]. There is a lower subsequent production of IL-12, IFNγand CD40-L, which also leads to a lower production of immunoglobulins compromising bacterial clearance [15,16]. The production of cytokines that contribute to Th17-cell polarization appears higher in neonates, and additionally elevated IL-10 production in early life was shown to be predominant in

B. pertussis infection cases [7,8]. Experimental studies in neonatal mice infected with B. pertussis showed lower lung inflammation but a higher systemic pathogenesis and a lack of upregulation of type I/III IFN responses, in contrast to adult mice [17,18]. These findings from animal models regarding a specific neonatal immune response to B. pertussis require confirmation in humans. Furthermore, previous studies employed immunological analyses restricted to certain cell types or cytokines; therefore, an integrative approach comparing adult and neonatal responses may offer novel insight.

In this study, we aimed to identify a specific neonatal systemic immune response profile to B. pertussis, through a comprehensive integration of immune cell phenotyping, cytokine measurement and transcriptomic analyses from an ex vivo human blood infection model.

## Results

We employed a multi-omics approach to characterize the neonatal immune response to B. pertussis.

A schematic overview is displayed in **Fig 1**.

### B. pertussis infection triggers a strong pro-inflammatory cytokine release signature

We first compared secreted levels of 27 cytokines and chemokines between non-stimulated (control, CTRL) and stimulated CB and AB with the B. pertussis FR4930 isolate (**Fig 2A** and S1 Dataset). Hierarchical clustering analyses of the cytokines and stratification by groups (CTRL, stimulated AB, stimulated CB) revealed clusters of elevated immune and inflammatory markers (**S1 Fig**). Among the 27 analytes, 15 were significantly elevated cytokines (adjusted p-value<0.05) in stimulated conditions (CB and AB) compared to CTRL (**Fig 2**). At baseline, ICAM levels were elevated, as previously described [19]. When cytokine concentrations from stimulated CB and AB were directly compared, 10 showed differential secretion (**Fig 2B**); of these, five were significantly upregulated upon stimulation: VEGF-A was found more elevated in AB whereas levels of IL-1α,IL-1β, IL-12p40 and TNF-α were higher in CB after stimulation with B. pertussis [19,20]. IFNγ levels were higher in CB than in AB, but the stimulation-specific effect did not reach statistical significance, likely due to elevated baseline IFNγ levels and inter-donor variability. Above B. pertussis concentrations of $1.75 \times 10^7$ CFU/mL, little additional effect on cytokine production was observed under our experimental conditions (**Fig 2C**). Of note, cell death rates were comparable between different B. pertussis concentrations (i.e., from 0 to $4.2 \times 10^8$ CFU/mL, all medians lower than 30%, **S1B Fig**). In summary, inflammatory cytokines were detected in both CB and AB after stimulation with B. pertussis, however, specific key immune activators were observed at significantly higher levels in CB.

We then investigated secreted cytokine/chemokine levels using a more targeted 19 analyte-panel (**S2 Fig**) in CB samples (n=24) stimulated with different B. pertussis bacterial strains (Tohama, and circulating isolates FR5730, FR6440, FR4930, FR5333, FR5862, range from $2.06 \times 10^7$ to $3.5 \times 10^8$ CFU/mL). Non-stimulated whole blood supernatants were used as CTRL. High levels of cytokines/chemokines were observed in CB stimulated with all B. pertussis isolates (**S2A Fig**). No specific significant profile was observed between isolates (Tohama strains and circulating isolates). To analyze whether this cytokine response is specific to B. pertussis infection and whether it is strain-dependent, we investigated effect of E. coli stimulation (E. coli S88 strain) on human cord blood. The E. coli strain was used as a reference, as it is a Gram-negative bacterium often responsible for severe infections in young infants. A range of E. coli concentrations was used for CB stimulation, selecting those that resulted in cell mortality levels comparable to those induced by B. pertussis

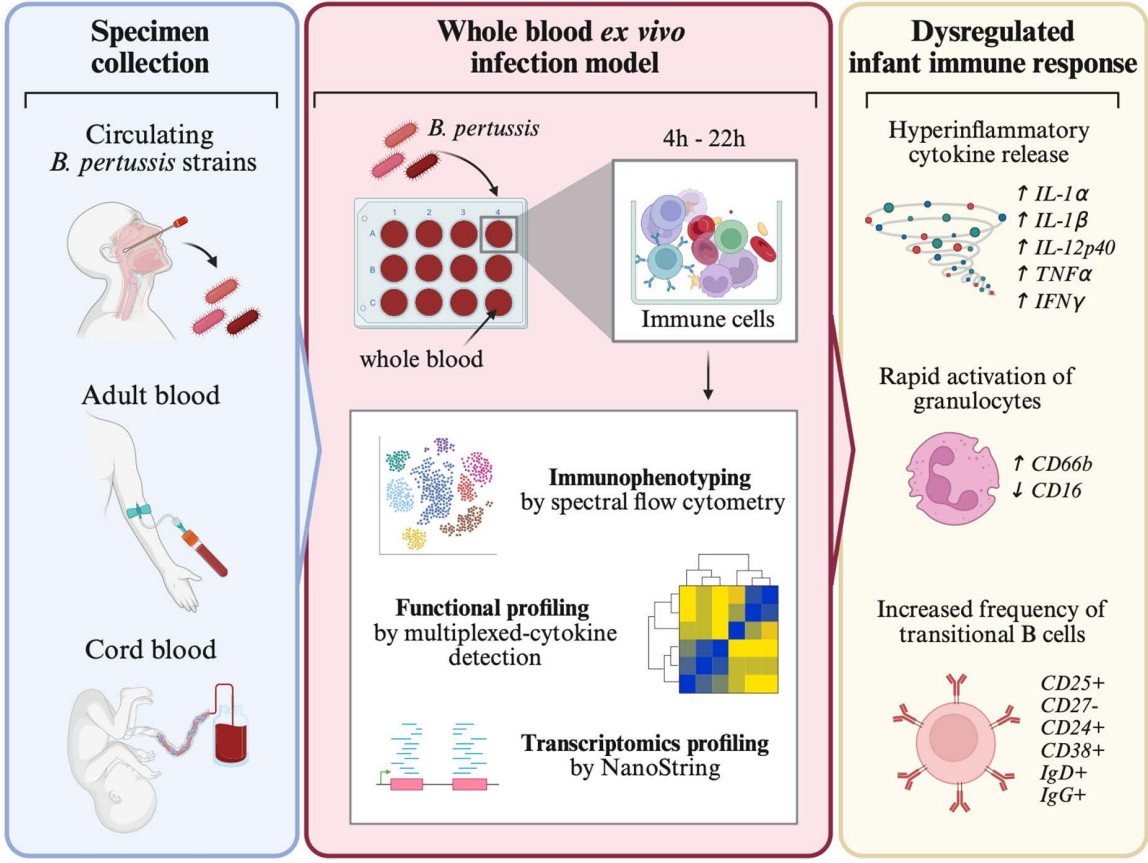

**Fig 1. Experimental designs and readouts.** Adult blood (AB) and cord blood (CB) samples were stimulated by different *B. pertussis* isolates (FR4930, FR5730, FR5862, FR5333, FR6440 or Tohama) at concentrations ranging from $1.75 \times 10^7$ to $4.2 \times 10^8$ CFU/mL for 4h or 22h. Cytokine release was analyzed using Luminex assays, immune cell profiling was performed by flow cytometry, and transcriptomic analysis was conducted by NanoString Technologies. Figure created in BioRender. Sadi, M. (2025) https://urldefense.com/v3/__https://BioRender.com/ctfvluo__;!!JFdNOqOXpB6UZW0!s9oPd_cZWW4YU9Tj_YyjedOUqXppKnb2gOy6VkBy9GgZkyi2Wl5433Uko5wi9ise2apVl_HrqzyrJ8vuKkTAxFg$.

(i.e., < 30% cell mortality; concentrations from $1.93 \times 10^6$ to $6.725 \times 10^7$ CFU/mL; **S2C Fig**) and still triggered an inflammatory response (**S2D Fig**). Stimulation with *B. pertussis* induced a stronger pro-inflammatory profile compared to *E. coli* (**S2B Fig**): among the 19 cytokines measured, 10 were significantly more elevated in CB stimulated by *B. pertussis* (adjusted p-value < 0.05), including TNF-α, IL-1α and IL-12p40, and other proteins, such as S100A8 and L-selectin. This suggests a specific immune response in CB to *B. pertussis* exposure.

### *B. pertussis* induced rapid activation of granulocytes, loss of detection of mMDSCs and B cell activation and differentiation

We then investigated changes in 13 immune cell populations following *B. pertussis ex vivo* stimulation using multiparametric flow cytometry analyses (**Fig 3A**, flow gating strategies and cell population identifications are described in **S3 Fig**). CB at the basal state, showed enriched populations of erythroid progenitors and monocytic myeloid-derived suppressor cells (mMDSCs) as compared to AB (**Fig 3B**). Following 4-hour stimulation, granulocytes displayed a phenotype characterized by increased CD66b expression associated with a loss of CD16 surface proteins in AB and CB that could be consistent with neutrophil degranulation or early apoptosis (**Fig 3C**). We also observed a loss of detection of mMDSCs in CB

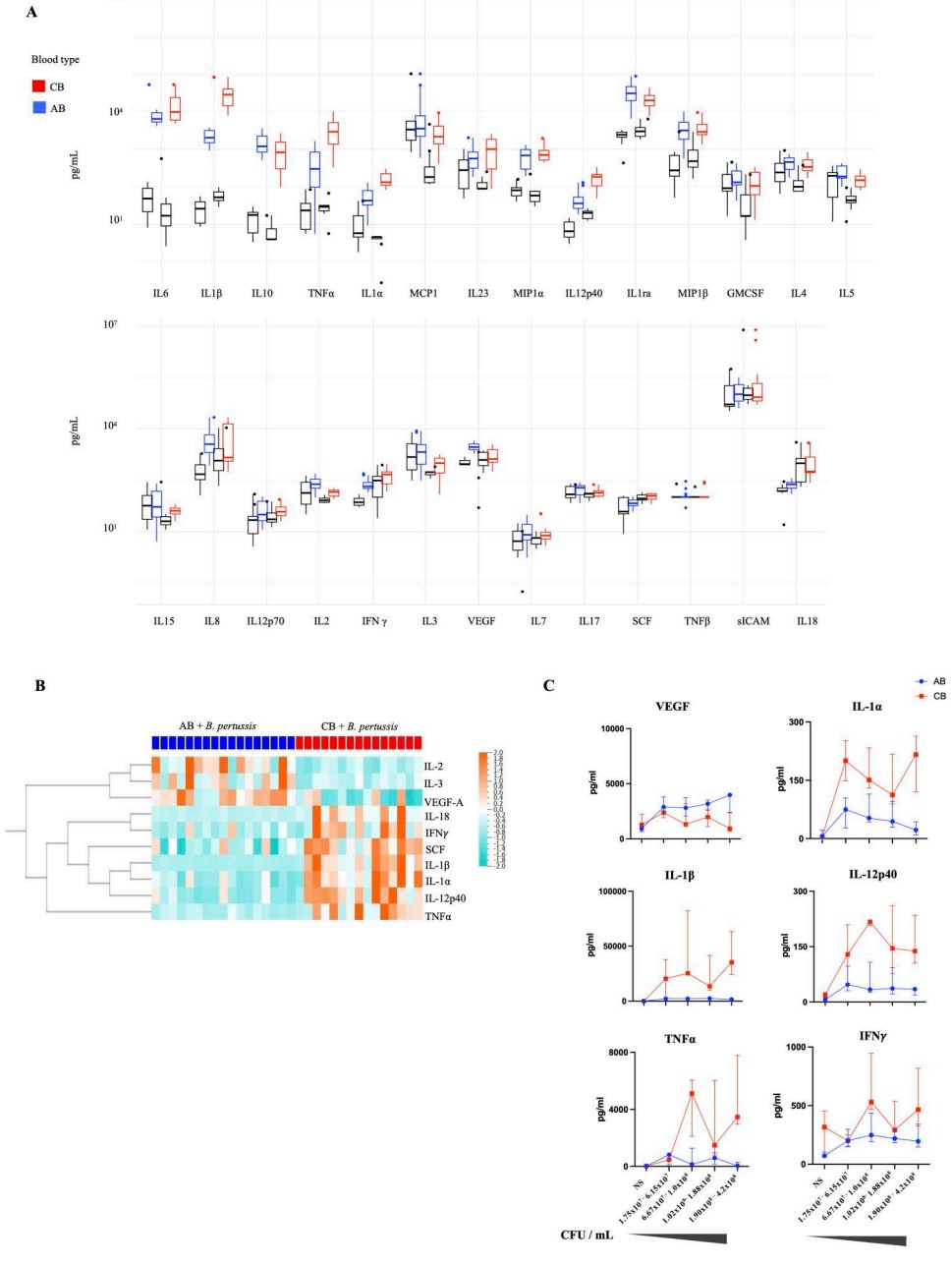

**Fig 2. Analyses of cytokine/chemokine levels in plasma samples.** Cord blood (CB, n = 8) and adult blood (AB, n = 9) from healthy donors were stimulated *ex vivo* with *B. pertussis* (isolate FR4930, $1.75 \times 10^7$ to $4.2 \times 10^8$ CFU/mL). Analytes were measured in whole blood plasma/supernatant after 22 hours of infection using a 27-analytes panel (Human Luminex Assay R&D Systems, Minneapolis). **(A)** Raw data of concentrations (in pg/ml) of 27 cytokines, chemokines and growth factors measured on plasma from AB (blue plots) and CB (red plots) stimulated with *B. pertussis* (FR4930) for 22 hours. Non stimulated plasma samples were used as controls (in black). Levels are represented in median and interquartile. **(B)** The heatmap displayed all analytes differentially secreted by stimulated CB (in red) compared to stimulated AB (blue) samples (adjusted p-value < 0.05). In heatmap **(B)** cytokines/chemokines are expressed in pg/mL and log transformed, with blue to red colors representing lower to higher secretion respectively. On the x-axis, experiments are organized by blood type (blue, AB; red, CB) and by concentration levels of *B. pertussis* (left to right, from $1.75 \times 10^7$ to $4.2 \times 10^8$/ml CFU/mL). On the y-axis, cytokines/chemokines are displayed following hierarchical clustering. Heatmap were created using Qlucore OMICS explorer 3.7. **(C)** Levels (in pg/mL) of VEGF-A, IL-1α, IL-1β, IL-12p40, TNFα and IFNγ in AB (blue) and CB (red) are displayed according to the levels of *B. pertussis* used for stimulation (from 0 to $4.2 \times 10^8$ CFU/mL). Each dot/square represents the median level of the cytokine according to the appropriate range of CFU/mL with its interquartile.

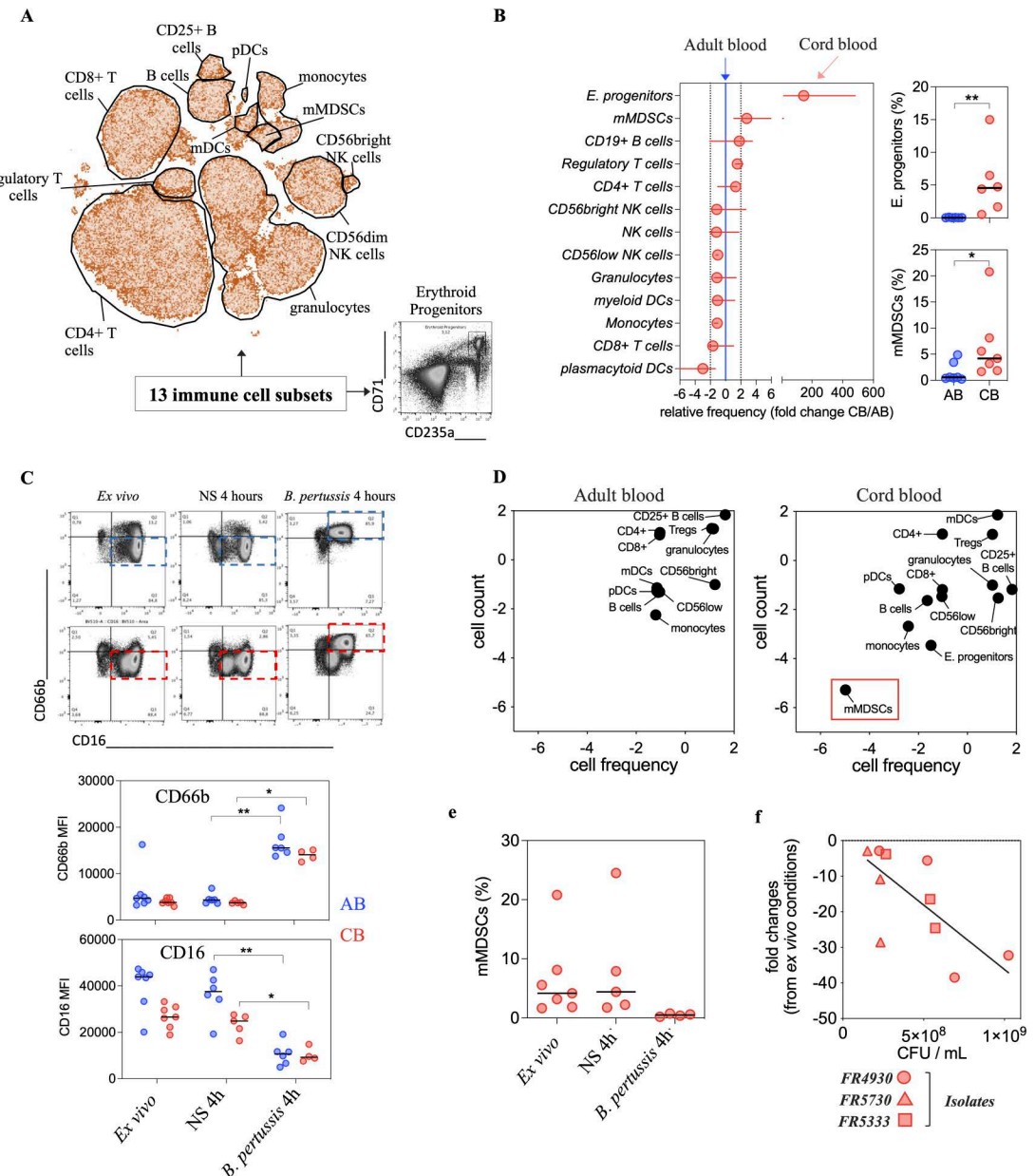

**Fig 3. Characterization of cell populations in CB at baseline and after a 4-hour stimulation with *B. pertussis*. (A)** Concatenated t-distributed stochastic neighbor embedding (t-SNE) analysis of CB samples to identify 12 distinct immune cell types based on the expression of phenotypical markers by spectral flow-cytometry. Erythroid progenitors were added to the analysis in a separated spectral flow-cytometry panel. **(B)** Relative frequency distribution of the different immune cell types at baseline in CB as compared to AB. **(C)** Representative flow cytometry plots of CD66b and CD16 expression by granulocytes from whole-blood, *ex-vivo* or after 4h at 36°C, 5% $CO_2$ in the absence (non-stimulated, NS) or presence of *B. pertussis* (isolate FR4930). Dot-plot representation (with median) of the Mean Fluorescence Intensity (MFI) for the expression of CD66b and CD16, across the different experimental stimulation conditions, as well as across CB (N = 4) and AB (N = 6). **(D)** Correlation analysis to assess the relationship between cell count and cell frequency across the different immune cell types from AB and CB, after 4-hour stimulation with *B. pertussis* (FR4930) and as compared to baseline (NS). Each point represents the median of relative cell count and cell frequency for the different immune cell types. **(E)** Dot plot histograms displaying the cell frequency of mMDSCs in CB across the different experimental conditions. **(F)** Linear regression analysis to model the relationship between the loss of detection of mMDSCs (expressed as Fold Changes as compared to *ex-vivo*) and the inoculum concentration of different *B. pertussis* isolates (FR4930, FR5730 and FR5333, expressed in CFU/mL).

stimulated with *B. pertussis* (**Fig 3D** and **3E**). This phenomenon was independent of the type of *B. pertussis* isolate used for stimulation but was correlated with its concentrations (**Fig 3F**). Importantly, mMDSCs did not appear within the pool of dead cells, suggesting phenotypic differentiation (S4 Fig).

After longer stimulation (22 hours) with *B. pertussis*, we identified a decrease in cell frequency and cell counts of monocytes in both AB and CB (**Fig 4A**). We also observed a strong increase in CD25+B cells in both AB and CB (**Fig 4A**, spectral cytometry plots represented in **S5A Fig**). These results obtained in whole blood, were confirmed in PBMCs (**S5B Fig**). We obtained consistent results across multiple clinical *B. pertussis* isolates (FR4930, FR5730 and FR5333) as well as with the reference strain Tohama, but not after stimulation with the *E. coli* (S88) isolate (**Fig 4B**). These results suggest a specific cellular response to *B. pertussis* that is independent of blood type. Analysis of these CD25+B cells in CB revealed a CD27-CD38+phenotype, with membrane expression of IgD, IgG, CD24, CD40 and CD80, compared to CD25-negative B cells (**Fig 4C**). This phenotype suggests the differentiation of B cells towards an activated transitional B cell phenotype [21,22].

### Transcriptomic profiles of whole CB and AB upon 4-hour stimulation with *B. pertussis*

To investigate gene expression patterns involved in systemic immune responses to *B. pertussis*, we analyzed the expression of 793 genes in stimulated CB and AB samples, comparing them to their respective non-stimulated controls (CTRL). Principal component analysis (PCA) revealed that samples from stimulated CB clustered separately from stimulated AB (**S6 Fig**). Investigating gene expression upon stimulation in CB, 54 genes were upregulated ($\log_2$FC > 1.5, adjusted p-value < 0.05) and 5 genes were downregulated ($\log_2$FC < -1.5, adjusted p-value < 0.05) after a 4-hour stimulation (**Fig 5A**). Pathway enrichment analysis revealed enhanced expression (-logit p-value < 5) of genes involved in interferon response, chemokine signaling and TNF signaling, among others (**Fig 5B**). Genes with the highest $\log_2$FC (> 2.5) were genes involved in chemotaxis and cell migration or myeloid activation (*CCL2, CCL18, CCL20, CCL3, CCL4, CXCL2, CXCL3* and *CXCL10*), IL-1 signaling or Th17 differentiation (*IL1α, IL1β, IL12β,IL23A*) (S2 Dataset). These highly upregulated genes were also identified in AB upon stimulation (S2 Dataset). We then explored differential gene expression between AB and CB upon a 4-hour stimulation (**Fig 6A**). Interestingly, several genes were upregulated in AB and not in CB. These were genes involved in the IFN signature such as *CSF3, CTSL* (**Fig 6B**), and adaptive immune responses related to JAK-STAT signaling such as *IL13* and *IL4R* (**Fig 6B**). *TREM1* was also found to be differentially elevated in AB. Furthermore, many genes downregulated in AB following *B. pertussis* stimulation, were still highly expressed in CB (**Fig 6C**); These genes are mainly involved in the activation of the innate immune response and cell migration, such as *TLR9, MAPK14, S100A8, S100A9* and *CEACAM1* (**Fig 6C**). In summary, gene expression in AB indicated activation of the adaptive immune response, particularly pathways involving myeloid activation, while innate immune responses were enriched in CB (**Fig 6D**).

### Transcriptomic signature of CD25+B cells upon 22-hour stimulation of CB with *B. pertussis*

CD25+B cells were found to expand in whole blood following *B. pertussis* stimulation. To identify genes specifically expressed in these B cells, we sorted them by flow cytometry based on their CD25 expression and performed differential gene expression analysis. This analysis compared the CD25+B cell subset after 22-hour stimulation of CB with *B. pertussis* (strain FR4930) with a pool of non-stimulated B cells (**Fig 7** and S2 Dataset). We identified 17 upregulated genes ($\log_2$FC > 1.5, adjusted p-value < 0.05) and one downregulated gene (NLRP1, $\log_2$FC < -1.5, adjusted p-value < 0.05) in stimulated CD25+B cells. Apart from the expected expression of *CD25* (*IL2RA*), the five genes with the highest $\log_2$FC (> 2.5) were involved in chemokine signaling and migration (*CCL2, CXCL3, CXCL5*), and activation markers (*IL6, IL1β*). The upregulation of *CD80*, another activation marker, corroborated findings obtained through flow cytometry. Other genes found to be highly expressed in these CD25+B cells are involved in proliferation and cell survival of B cells as well as T cell regulation (e.g., *IL21R, CCL22*) [23,24], immune cell crosstalk and immunosuppression (e.g., *CD274, IDO1, EBI3*)

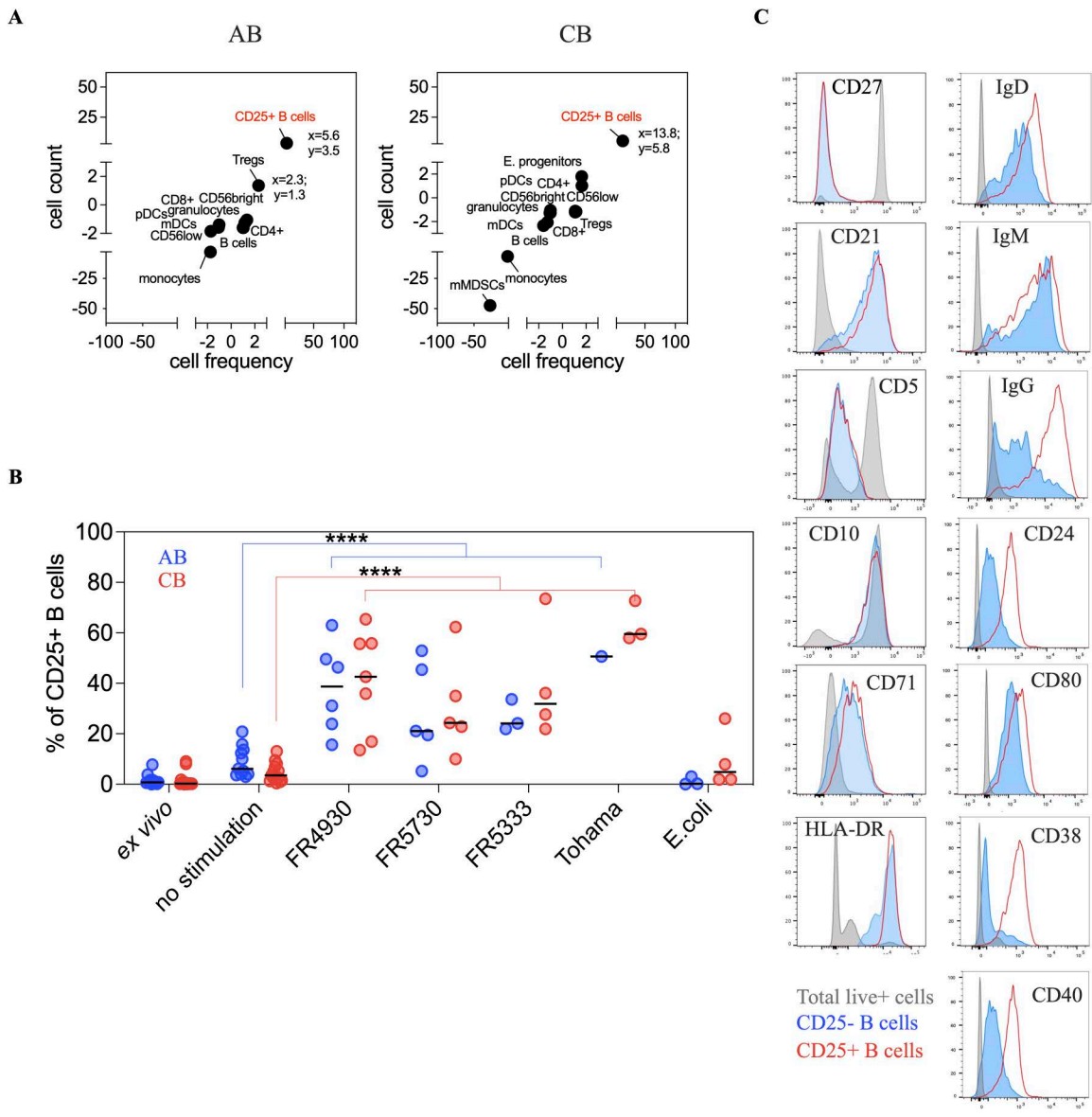

**Fig 4. Characterization of cell populations in CB after a 22-hour stimulation with *B. pertussis*. (A)** Correlation analysis to model the relationship between cell count and cell frequency across the different immune cell types from AB and CB, after 22h stimulation with *B. pertussis* (FR4930). Each point represents the median of relative cell count and cell frequency for the different immune cell types after stimulation, when compared to base-line control (22-hour incubation NS). **(B)** Dot plots showing the frequency of CD25+B cells in adult blood (AB, blue) and cord blood (CB, red) across experimental conditions. Four *B. pertussis* isolates (FR4930, FR5730, FR5333, and Tohama) and a reference *E. coli* strain (S88) were tested. Sample sizes (AB/CB): ex vivo (11/14), non-stimulated (11/14), FR4930 (6/7), FR5730 (5/5), FR5333 (3/4), Tohama (1/3), *E. coli* (3/4). For statistical analyses, all *B. pertussis*-stimulated samples were pooled into a single group (AB, n = 18; CB, n = 23), regardless of isolate. **(C)** After 22 hours of stimulation with *B. pertussis* (FR4930 isolate), cells were all stained for CD3, CD19, CD25 and for either CD27, CD21, CD5, CD10, CD71 (1st B-cell panel), or for CD24, CD38, CD40, CD80, IgD, IgM and IgG (2nd B-cell panel), and analyzed by flow cytometry. CD19+CD25-negative cells were represented in blue, CD19+CD25+cells were represented in red, and total live cells in grey.

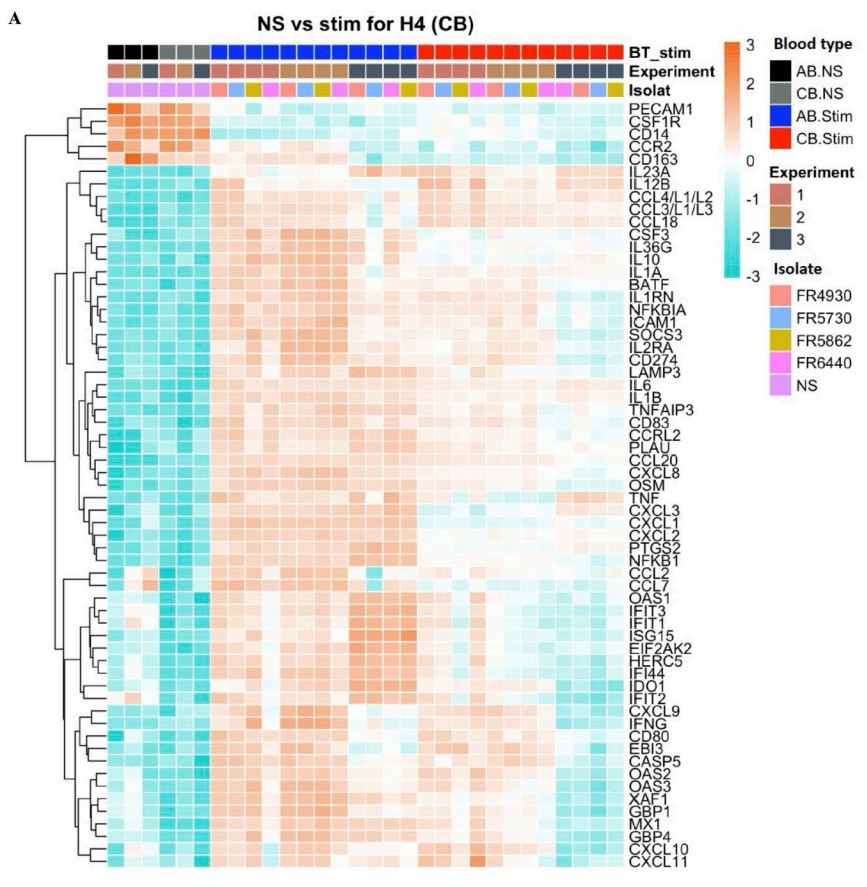

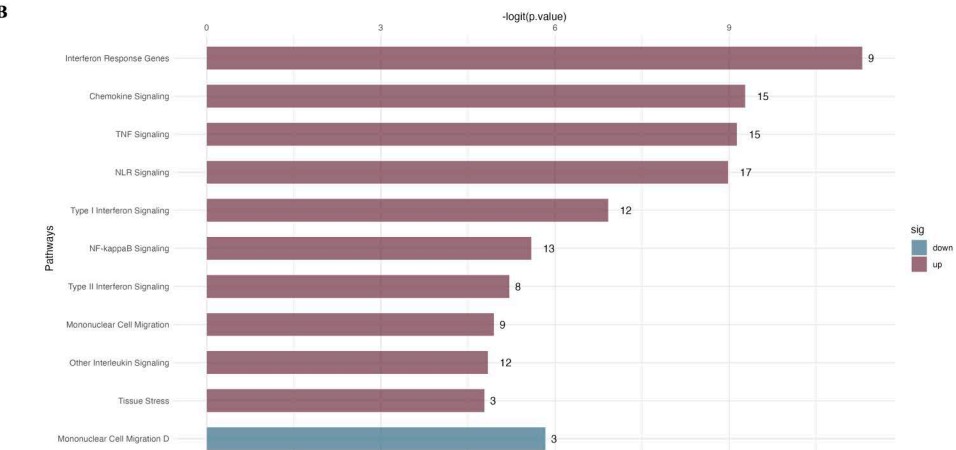

**Fig 5. Genes differentially regulated in CB after a 4-hour stimulation with *B. pertussis*. (A)** Heatmap representing differentially expressed genes (log$_2$FC ± 1.5, adjusted p-value < 0.05) in cord blood (CB) stimulated *ex vivo* with different *B. pertussis* isolates (FR4930, FR5730, FR5862, FR6440) compared to NS CB conditions. On the x-axis, blood donors are organized by blood type, the experiment (n = 3 independent experiments, corresponding to 3 different donors for each blood type) and the *B. pertussis* isolate used for *ex vivo* stimulation. The y-axis corresponds to each gene displayed following hierarchical clustering. The heatmap visualizes normalized gene expression values scaled per gene by centering the mean and dividing by the standard deviation, providing an overview of gene expression patterns. Blue to red colors represent lower to higher expression respectively. **(B)** Gene expression pathways enriched in CB stimulated with *B. pertussis*. The bar plots depict pathways that are significantly enriched (dark red, -logit(p-value) > 2.5) or diminished (blue). Pathways are listed on the y-axis, while the x-axis represents the corresponding -logit(p-value).

PLOS Pathogens

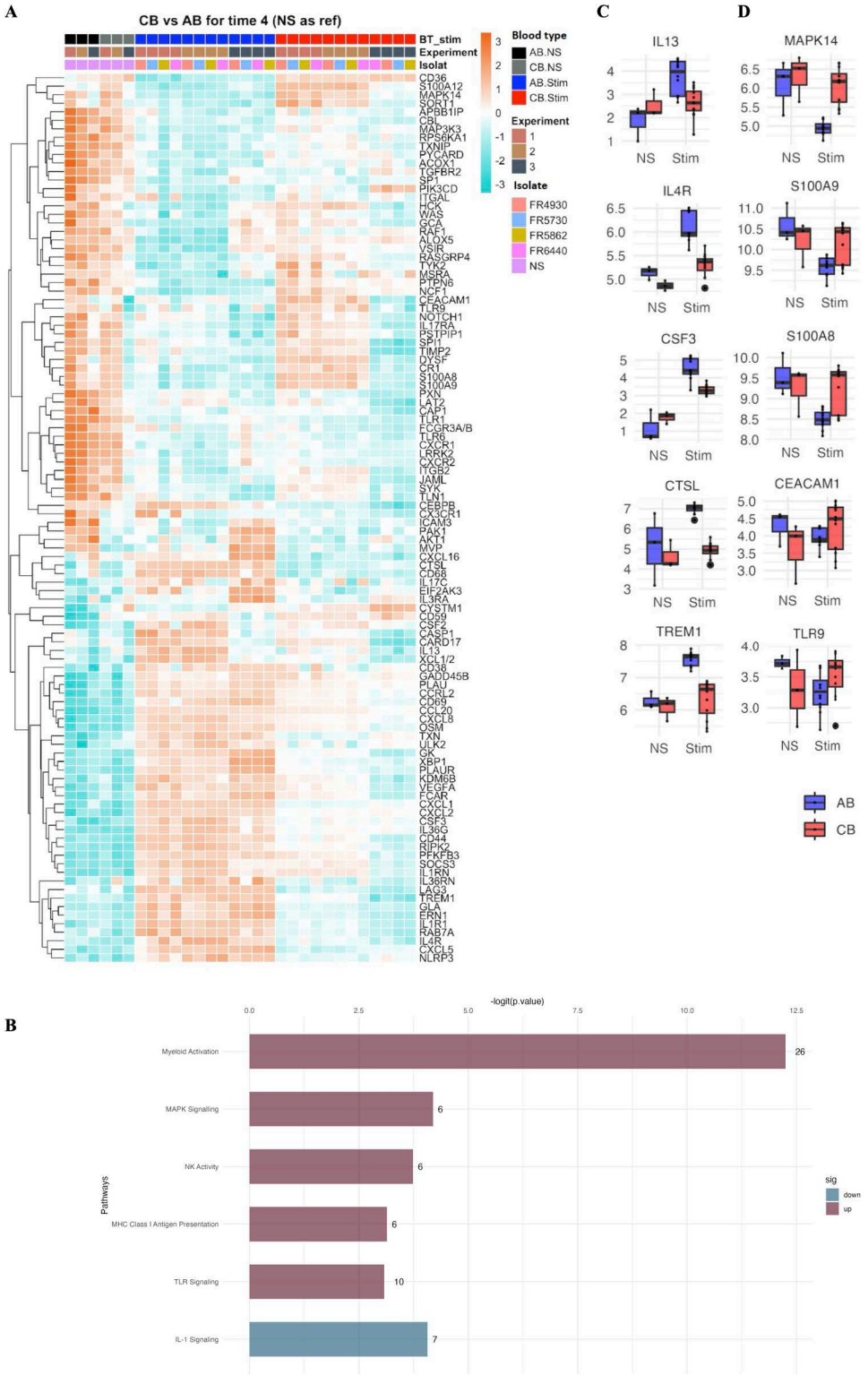

**Fig 6. Genes differentially regulated between CB and AB after a 4-hour stimulation with _B. pertussis_. (A)** Heatmap representing the genes with differential slopes (log₂FC=0, adjusted p-value<0.05) between AB and CB, where each slopes reflects the expression trajectory between no stimulation (NS) and stimulation conditions. On the x-axis, blood donors are organized by blood type, the experiment (n=3 independent experiments corresponding to 3 different donors for each blood type) and the _B. pertussis_ isolate used for _ex vivo_ stimulation. The y-axis corresponds to each gene displayed

following hierarchical clustering. The heatmap visualizes normalized gene expression values scaled per gene by centering the mean and dividing by the standard deviation, providing an overview of gene expression patterns. Blue to red colors represent lower to higher expression respectively. (**B**) Boxplots representing upregulated genes in AB, as compared with CB (changes in gene expression in response to infection were compared, adjusted p-value < 0.05). (**C**) Boxplots representing downregulated genes in AB, compared to CB (adjusted p-value < 0.05). (**D**) Gene expression pathways enriched in CB compared to AB, upon stimulation with *B. pertussis*. The bar plots depict pathways that are significantly enriched (dark red, -logit(p-value) > 2.5) or diminished (blue). Pathways are listed on the y-axis, while the x-axis represents the corresponding -logit(p-value).

[25] (**Fig 7**). Taken together, these results suggest that the CD25 + B cells identified in whole blood following *B. pertussis* stimulation are activated B cells that exhibit an immunoregulatory gene profile.

## Discussion

The most severe forms of pertussis in neonates present as a multiorgan disease alongside the common respiratory tract infection, suggesting that the underlying pathogenesis in neonates differs markedly from that in adults, with a specific systemic immune response [5,17]. In this study, an integrative analysis of the early immune responses in human CB and AB samples stimulated with *B. pertussis* isolates, revealed an inflammatory profile specific to CB, accompanied by an early loss of mMDSCs and concurrent upregulation of genes involved in myeloid activation and innate immune responses. Additionally, *B. pertussis* induced B cell remodeling characterized by an increased CD25 + B lymphocyte fraction in both blood types. In CB, these CD25 + B cells exhibited markers specific for activated transitional B-cells, along with the expression of various immunomodulatory genes.

In an *ex vivo* model of whole blood infection with *B. pertussis*, we detected inflammatory markers in the supernatants of both blood types. However, the levels of IL-1α, IL-1β, IL-12p40, TNF − α were significantly higher in CB. These cytokines are all involved in inflammasome and Th17 immune responses [26–28]. This has been observed in previous cord blood models in response to whole bacteria or purified bacterial effectors [10]. This may be explained by an LPS specific

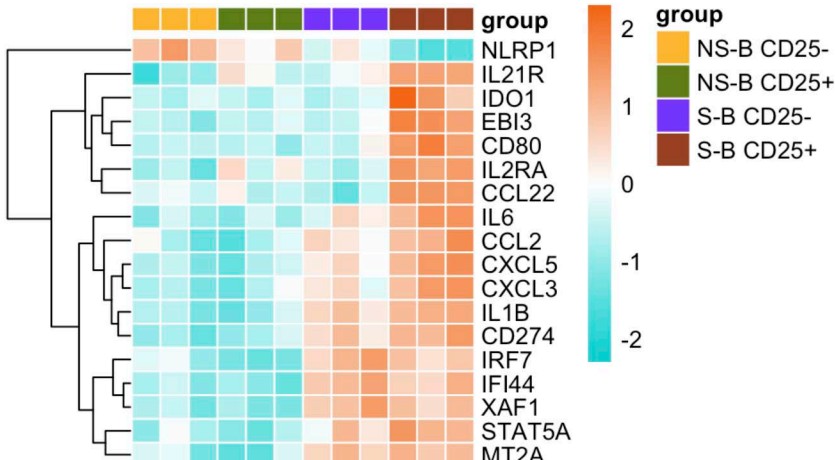

**Fig 7. Gene expression in isolated B cells derived from CB.** Heatmap representing differentially expressed genes (log$_2$FC ± 1.5, adjusted p-value > 0.05) in B cells upon stimulation with *B. pertussis* FR4930. CD25 + B cells stimulated for 22h are compared to non-stimulated CD25+ and CD25- B cells. On the x-axis, the isolated B cells are organized according to stimulated (S) or non-stimulated (NS) CD25- or CD25 + B cells. The data is representative of three independent experiments. The y-axis corresponds to each gene displayed following hierarchical clustering. The heatmap visualizes normalized gene expression values scaled per gene by centering the mean and dividing by the standard deviation, providing an overview of gene expression patterns. Blue to red colors represent lower to higher expression respectively.

hyperinflammatory response [29]; however, it has not been observed following stimulation with *E. coli*, which may indicate a more specific response to *B. pertussis*. As lower CFU levels of *E. coli* were required to achieve comparable levels of cell death, which may introduce bias, further studies using *B. pertussis* LPS for stimulation will be needed to confirm our results. These findings suggest a pro-inflammatory imbalance in CB, which might reflect a potential uncontrolled deleterious response in neonates, as observed in mouse models of sepsis and septic shock [30,31].

The cellular phenotypes identified during early infection (4-hour stimulation with *B. pertussis*) might participate to the hyperinflammatory state observed in CB. Notably, our CB samples exhibited a significant reduction in mMDSCs following *ex vivo* infection with *B. pertussis*. Interestingly, Gervassi *et al*. showed that neonatal blood at baseline contains MDSCs, which suppress CD4+ and CD8+ T cell proliferative responses *ex vivo* [32]. This finding underscores the pivotal role for these cells, along with CD71+ progenitor cells, in maintaining immune homeostasis during early human life [10,32], and regulating inflammation during infection [33]. Supporting the hyperinflammatory phenotype, granulocytes showed rapid activation with increased CD66b expression and a loss of CD16 surface proteins in AB and CB. This phenomenon, observed in different contexts of septic shock [34–36], and has been associated with impaired neutrophil functions and heightened inflammation [37]. Also, the reduced CD16 surface expression observed after *B. pertussis* stimulation suggest neutrophil degranulation or early apoptosis, rather than classical activation pathways [38,39].

In the context of severe pertussis, an increase of naïve B and T cell proliferation has been documented, while the underlying mechanisms remains elusive [40]. In our model at a later stage of infection (22-hour stimulation with *B. pertussis*) we observed an increased fraction of CD25+B cells. Compared to CD25- B cells, these cells exhibited additional markers of activation (CD40), and displayed a memory transitional phenotype (CD27-CD38+CD24+IgD+IgG+) [7]. Several studies pinpointed the suppressive role of transitional B cells on naïve T cell differentiation into T helper (Th) type 1 or Th17 in part through the production of IL-10 [21,41]. Functional analyses of CD25+B cells, including cytokine production and capacity to affect T cells, might be of interest to better characterize their functions in the context of pertussis disease.

Transcriptomic analyses of CB and AB after 4 hours of stimulation indicated an upregulation of several chemokine genes involved in cell migration. These systemic chemotactic signals, implicated in the leukocytosis observed in severe pertussis disease, may result from the expression of pertussis toxin, a key virulence factor of *B. pertussis* [40]. We also observed a strong innate immune response with similarly high expression of *IL-1β,IL-1α, IL-12p40, TNF−α*, and *IFN-γ* genes in both blood types. However, at the protein level, levels of these cytokines were significantly more elevated in CB possibly due to post-transcriptional regulation or rapid gene downregulation in AB [42]. This discrepancy may indicate the capacity of the AB immune system to mount a more controlled response to infection. Notably, unlike in AB, CB showed no gene downregulation upon stimulation but sustained high expression of innate immune genes (*MAPK14*, *S100A8*, *S100A9*, *CEACAM1* and *TLR9*). S100A8 protein was also detected in the plasma after 22-hour stimulation. Alarmins S100A8 and S100A9, rapidly released from myeloid cells under stress, act as amplifiers of inflammation during infection, but also seem to be physiologically elevated at birth [43]. S100 proteins might also alter monocyte phenotypes in neonates and could have contributed to the mMDSC loss that we observed [44]. Elevated CEACAM1 serum levels have been linked to T-cell suppression in neonatal sepsis [45]. These proteins might be considered for future research, as they could be potential therapeutic targets for the treatment of severe pertussis.

Transcriptomic analysis of stimulated CD25+B cells derived from stimulated CB revealed an activated profile when compared to non-stimulated B cells. Also, *IL21R,* involved in B cell fate [24], is known to be expressed on transitional B cells [46,47]. Its upregulation further reinforces that the CD25+B cell subset observed after *B. pertussis* stimulation, retains an immature state, as indicated by the cellular phenotype identified through flow cytometry. Interestingly, we also identified upregulated genes involved in immunoregulation, such as *CD274* (*PD-L1*). CD274 has previously been associated with CD4+ and CD8+T cell suppression *ex vivo* [48]. Upon *B. pertussis* infection, CD274 is upregulated on human monocyte-derived dendritic cells, suggesting pathogen-induced immunosuppression [47]. Additionally, genes encoding other immunomodulatory proteins, including EBI3, XAF1, IFI44 and IDO1, were found to be upregulated in

CD25 + stimulated B cells. Some of these genes have been associated with regulatory B cell function in sepsis, possibly indicating regulation of the adaptive immune response to prevent aberrant activation [25,49–51]. IDO1, a tryptophan converting enzyme, has been shown to be upregulated during *B. pertussis* infection in infant mice compared to adult mice [51]. It was hypothesized that elevated IDO1 levels may lead to tryptophan depletion, promoting immunosuppression and contributing to severe pertussis disease. Moreover, CXCL3 and CXCL5 chemokine genes were upregulated in stimulated CD25 + B cells, compared to non-stimulated CD19 + B cells. CXCL3 is a neutrophil attractant [52,53], while CXCL5 inhibits their recruitment into the lung, impairing bacterial clearance and reducing survival in a mouse *E. coli* pneumonia model [54]. CXCL5-mediated prevention of neutrophil recruitment to the site of infection might contribute to leukocytosis in severe pertussis disease. We also observed the downregulation of the inflammasome-associated *NLRP1* gene in the stimulated CD25 + B cell subset compared to non-stimulated CD19 + B cells, which further supports a regulatory function [55]. Taken together, we identified a distinct activation-associated transcriptomic profile with upregulation of maturation enhancers and immunosuppressors in stimulated CD25 + B cells. Further studies are needed to assess whether this, along with CD25 + B cell expansion, reflects impaired adaptive immunity and contributes to pertussis pathogenesis.

One limitation of this study is the *ex vivo* nature of the model. *B. pertussis* infection typically begins as a respiratory tract infection. While some experimental and human studies have reported systemic circulation of the bacteria [17,56,57], the results of this model cannot be fully extrapolated to an actual *B. pertussis* infection. A translational study on young patients infected by *B. pertussis* at different ages and clinical severity will be necessary to confirm these results *in situ*. Also, our study lacks information on vaccination history of the donors, as both CB and AB samples were obtained from anonymous individuals. Although the negative serology suggests an absence of recent natural exposure or vaccine-induced immunization, previous vaccinations could still have influenced immune responses, particularly B and T cell–mediated ones. Future studies including detailed immunization or comparing cohorts with different vaccine backgrounds would help clarify these effects on maternal and cord blood immunity. Another limitation is that cytokine measurements were based on released cytokines from whole blood stimulations, reflecting the global secretome following stimulation with *B. pertussis* rather than the contribution of individual cell subsets. We further explored B cells, whose proportion markedly increased following stimulation, through transcriptomic analyses; however, extending this to all immune cell types was beyond the scope of the present study. Future work using approaches such as intracellular flow cytometry or single-cell analyses would provide valuable complementary information on the specific cellular sources of cytokines. Also, we do not fully explain the different gene expression levels in whole blood, as many cells are involved. To gain further insight into the mechanism behind *B. pertussis* blood cell infection, differentially expressed genes could be studied using single-cell RNA sequencing. Lastly, we have not investigated the kinetics of gene expression levels and proteomic markers, which may have led to the omission of important stages of the immune response. Temporal analysis of gene expression and protein dynamics is crucial to understanding the progression of *B. pertussis* infection, as both the host immune response and bacterial adaptation can vary significantly over time.

In conclusion, using multi-omics approaches and a unique whole blood infection system, this study highlights new aspects of the neonatal immune response induced by *B. pertussis*, revealing novel molecular and cellular mechanisms that could be targeted for the treatment of severe pertussis disease.

## Materials and methods

### Ethics statement

All blood donors provided written informed consent, and all blood samples were processed anonymously. This study was approved by the Comité Ethique et Scientifique pour les Recherches, les Etudes et les Evaluations dans le domaine de la Santé (CESREES, ethics and scientific committee for research, Studies and evaluation in health, registration number 2428977b, 11/02/2021) and authorized by the Commission Nationale de l'Informatique et des Libertés (CNIL, French data protection authority, registration number DR-2021–216, 20/07/2021).

## Bacterial isolates selection and growth conditions

Five *B. pertussis* isolates (FR4930, FR5333, FR5862, FR5730, FR6440), representative of currently circulating isolates from the biocollection of the French National Reference Center (NRC) for whooping cough, along with the reference strain "Tohama", were used for this study. Their phenotype characteristics are described in **S1 Table**. All isolates were first grown at 36 +/-1°C for 72 hours on Bordet-Gengou Agar (BGA) and then subcultured in this medium for 24h. Subsequently, the bacteria were grown at 36 +/-1 °C in Stainer Scholte (SS) medium from an initial Optical Density at 650 nm ($OD_{650}$) adjusted to 0.1 (Biomate 3S, Thermo Fisher Scientific, Waltham, MA), under agitation. The cultures were incubated for 24 hours to allow the bacteria to reach the log-phase, indicated by an $OD_{650}$ of 1 +/-0.1. Colony forming units (CFU)/mL were determined by serial dilution and spreading on BGA.

One *Escherichia coli* (*E. coli*) strain S88 isolated from a neonatal meningitis case was used as control [58]. *E. coli* S88 was first grown at 36 +/-1°C for 24 hours on Tryptic Soy Agar (TSA) plates and then subcultured in the same medium for 24 hours. Then, the bacteria were grown in 2 mL of Luria-Bertani broth, with the $OD_{600}$ adjusted to 0.1, under agitation and at 36 +/-1 °C. The culture was incubated for 3 hours to allow the bacteria to reach the log-phase, indicated by an $OD_{600}$ of 1 +/-0.1. CFU/mL were determined by serial dilution and cultured on TSA plates. At each stage, the absence of BGA and TSA contamination was controlled. Notably, the CFUs of the *E. coli* used for stimulation were lower than those of the *B. pertussis* due to a marked increase in cell mortality observed at concentrations exceeding $5x10^7$ CFU/ml.

### *Ex vivo* whole blood stimulation

Heparinized cord blood (CB) samples from healthy term neonates were collected by the cellular therapy department of Saint-Louis hospital (Paris, France) after informed consent. Blood samples from adult (AB) donors were obtained by the Etablissement Français du Sang (Paris, France) (**S2 Table**). Blood samples were all tested for anti-Pertussis toxin IgG antibody concentrations (kit Savyon SeroPertussis Toxin IgG 1231-01D, Savyon Diagnostics Ltd, Israel) to confirm their negativity (IgG titers <40 UI/mL).

The whole blood stimulation protocol was adapted from Jansen K et *al* [59]. Briefly, 200 µL for cytokines analyses and 500 µL for transcriptomics analysis of whole blood samples were incubated or not with various *B. pertussis* isolates or *E. coli* S88. After sampling few milliliters of each blood sample and red blood cell lysis, cells were counted following Trypan Blue Stain coloration (Biorad TC20). From an estimated concentration of $1.0x10^9$ CFU/ml of *B. pertussis* corresponding to a suspension of $OD_{650}$ = 1(laboratory data), we used the volume of bacteria necessary to reach an estimated MOI from 10 to 50 [60]. This corresponded to a measured CFU of $1.75x10^7$ to $4.2x10^8$/ml (**S3 Table**). For *E.coli* strain, we used concentrations from $1.93x10^6$ to $6.72x10^7$ CFU/mL, these concentrations were selected to obtain equivalent levels of cell death (<30%) (**S2C Fig**).Cell death was assessed either by Trypan Blue stain coloration using a TC20 automated cell counter (Bio-Rad) or by flow cytometry after staining with a fixable viability dye (Zombie NIR Fixable Viability Kit, BioLegend) following red blood cell lysis. Cell mortality was expressed as the percentage of non-viable cells (Trypan Blue–positive or Zombie NIR–positive) and reported as mean ± standard deviation from three independent experiments. Blood was mixed 1:1 with sterile pre-warmed (37°C) RPMI-1640 (RPMI, Invitrogen) + GlutaMAX-I (Gibco, Grand Island, NY, USA) on flat bottom well plates (Corning Falcon, NY 14831, USA). For multianalyte assays, whole blood was incubated with the microbial agent for 22 hours at 36 +/-1°C, supernatants were separated from cells by centrifugation (at 1200 rpm for 5 min at room temperature) and frozen at -80°C. For flow cytometry, blood samples were collected after 4- and 22-hours stimulation with *B. pertussis* isolates, for immediate processing and staining. Peripheral Blood Mononuclear cells (PBMC) from CB were also isolated with Histopaque-1077 Hybri-Max liquid (Sigma-Aldrich, Saint-Louis, Missouri, USA) and stimulated with *B. pertussis* for complementary flow cytometry analyses. For transcriptomic analyses, whole blood was incubated for 4 hours with *B. pertussis* isolates and then resuspended in 1.5 mL Trizol LS (Thermo Fisher Scientific), vortexed and incubated for 10 minutes at room temperature before -80°C storage.

## Multianalyte profiling

Concentrations of 27 cytokines, chemokines, and growth factors were measured in plasma from CB and AB using the Luminex-based multiplex bead technology (Human Magnetic Luminex Assays, R&D Systems, Minneapolis, MN, **S4A Table**). For further experiments in CB only, measurements focused on 19 analytes (Human Magnetic Luminex Assays, R&D Systems, **S4B Table**). All assays were conducted according to manufacturer's recommendations. Plates were measured using the Bio-Plex 200 System and analyzed with the Bio-Plex Manager software (version 6.0, Bio-Rad, Hercules, CA). The lower and the upper limit of quantification (LOQ) corresponded to the lowest and highest analyte concentration that can be quantified with acceptable accuracy and were determined based on the standard curve for each assay. The limit of detection is the lowest value read out after application of the standard curve and use of curve-fitting algorithms. Individual values below the lower LOQ (out of range <) were replaced with a value that is the lowest value measured in the data set divided by two; values upper the upper LOQ (out of range>) were replaced with a value that is twice the highest value measured in the data set.

## Flow cytometry

Whole blood samples were treated using red blood cell lysis buffer (BD Pharm Lyse, BD Biosciences, Franklin Lakes, NJ) for 10 min at room temperature. After centrifugation at 1500 rpm, cells were collected and incubated in the presence of Live-Dead fixable violet (Zombie NIR Fixable Viability Kit, Biolegend, San Diego, CA) according to the manufacturer's protocol.

For flow cytometry analysis of leukocyte subsets, the cells were washed with phosphate-buffered saline containing 2% FBS, pelleted and then stained with surface antibodies (see list of antibodies in **S5 Table**) in Cell Staining Buffer (Biolegend) for 30 min at 4°C. After two consecutive wash steps, cells were either fixed in a solution of 2% paraformaldehyde and analyzed on a spectral analyzer (SP6800, Sony Biotechnology, San Jose, CA), or directly analyzed on a FACS AriaFusion (BD Biosciences). Data were analyzed using FlowJo software (version 10.1r5).

For fluorescence-activated cell sorting (FACS), the cells marked with the Live-Dead fixable violet were centrifuged at 1500 rpm and 4°C. The supernatant was discarded, and the cells were stained with CD4, CD19 and CD25 antibodies (**S5 Table**) in Cell Staining buffer for 20 min at 4°C. Unbound antibodies were washed out, the pellet was resuspended in 500 µl of Cell Staining buffer and immediately chilled on ice. FACS was performed on a BD FACSymphony S6 Cell Sorter using the BD DIVA Software (BD Biosciences) and analyzed using the FlowJo software (version 10.1r5). The sorted cells were recovered in 200 µl of PBS BSA 0.5% and centrifuged for 8 min at 1900 rpm and 4°C. Depending on the cell count, the pellets were then either resuspended in RLT buffer (Qiagen) diluted 1:3 in nuclease-free water (<150,000 cells), or in undiluted RLT buffer (>150,000 cells) for subsequent NanoString analysis, following the manufacturer's protocol.

## RNA extraction and quality controls

Whole blood samples fixed in Trizol LS were thawed on ice 60 min and centrifugated. RNA extraction was performed on supernatants with the RNeasy mini kit (Qiagen Inc, German Town, MD) according to manufacturer's protocol. RNA was quantified using Qubit RNA HS Assay kit (Invitrogen, Thermo Fisher) and a Qubit 3.0 fluorometer (Thermo Fisher). A total of 100 ng RNA per sample is required for analysis with NanoString Technologies (Seattle, WA, USA). Quality control was performed using the 2100 Bioanalyzer with the RNA 6000 pico kit (Agilent Technologies, Santa Clara, CA) and the 2100 Expert software. RNA Integrity Number (RIN) scores greater than 6 were obtained for each sample before analysis.

## Gene expression analysis

The NanoString nCounter platform was used for digital RNA transcript counting according to manufacturer protocols. Briefly, the analysis involved hybridizing 100 ng extracted RNA from each sample. For sorted cells, whole cell lysates were

directly analyzed using either 4.5 µl of lysate (<150,000 cells) or 1.5 µl of lysate (>150,000 cells). We used the NanoString Host Response panel, containing 773 genes covering the host immune response to infectious diseases and 12 internal reference genes for data normalization [61]. We added 20 genes to this panel, selected for their involvement in the immune response to *B. pertussis* infection as previously described (**S6 Table**) [62,63]. After Codeset hybridization, the samples were loaded on a cartridge *via* the NanoString Prep Station and subsequently analyzed with the nCounter Digital Analyzer.

### Statistical analyses

Cytokine heatmaps were made and analyzed with Qlucore OMICS explorer 3.7 (version 1.1.463). Differential analytes with an adjusted p-value < 0.05 for Luminex assays were included in the heatmaps. In CB vs AB plasma comparisons, analytes that were significantly differentially secreted after stimulation with *B. pertussis,* were depicted as curves with the x-axis increasing concentrations of

*B. pertussis* (in CFU/mL) used for stimulation, and in y-axis analyte concentrations in pg/mL with GraphPad PRISM (Version 9).

Statistical tests for cellular comparisons were based on spectral flow cytometry datasets and were performed using GraphPad PRISM (Version 9). Nonparametric Mann-Whitney U test or Kruskall-Wallis test followed by post hoc multiple comparison Dunn's test, were used to compare cell population proportions or medians of each sample Mean Fluorescence Intensities (MFI).

Data obtained from NanoString assays were normalized using nSolver Advanced Analysis Software v4.0 (RRID:SCR_003420) according to manufacturer's instructions, which applies gNorm to select the House Keeping genes. A principal component analysis (PCA) was performed on the normalized gene expression data to explore overall variation in the dataset and identify clustering patterns. The analysis was conducted using the ade4 package (v1.7.22) in R. Differential expression analyses of all normalized NanoString datasets were performed using the limma R package (v3.52.1) [64]. In the generalized linear model (GLM) implemented in limma package, we included the blood type and the stimulation status as main effect and interactions between blood type and stimulation. The plate variable was also added in the model as a batch (experimental) effect. Heatmaps visualize normalized gene expression values, scaled per gene by centering the mean and dividing by the standard deviation, providing an overview of gene expression patterns.

Changes in gene expression in response to infection were compared between CB and AB using a set of contrast vectors defined within the limma framework. Differentially expressed genes were considered significant if the adjusted p-value was < 0.05 with Benjamini Hochberg's method. Results were visualized using heatmaps and boxplots. Boxplots were used to illustrate changes in gene expression levels between experimental groups. Pathway enrichment analysis was performed using a one-tailed Fisher's exact test based on the hypergeometric distribution.

## Supporting information

**S1 Fig. Cytokine/chemokine levels in plasma samples of AB and CB after stimulation with *B. pertussis* and cell death induced by *B. pertussis*.** (**A**) Cord blood (CB, n = 8) and adult blood (AB, n = 9) from healthy donors were stimulated *ex vivo* with *B. pertussis* (isolate FR4930, $1.75 \times 10^7$ to $4.2 \times 10^8$ CFU/mL). Analytes were measured in whole blood plasma/supernatant after 22 hours of infection using a 27-analytes panel (Human Luminex Assay R&D Systems, Minneapolis). In heatmaps, cytokines/ chemokines are expressed in pg/mL and log transformed, with blue to red colors representing lower to higher expression, respectively. On the x-axis, blood donors are organized by groups and by increasing concentrations of *B. pertussis* (CFU/mL), and on the y-axis, cytokines/chemokines are displayed following hierarchical clustering. Heatmaps were created using Qlucore OMICS explorer 3.7. The heatmap displayed all analytes secreted in non-stimulated (NS) whole blood control (CTRL) samples (both NS CB and AB, in black) and in stimulated AB (in blue)

and CB (in red) samples. (**B**) Cell death rate in AB (blue histograms) and CB (red histograms) after stimulation (FR4930) with a range of bacterial inoculum (in CFU/mL, x-axis). Cell death rate is represented in means and standard deviation of three experiments.
(DOCX)

**S2 Fig. Analyses of cytokine/chemokine profiles of CB specific to *B. pertussis*.** Cord blood (CB) (n = 24) from healthy donors were stimulated *ex vivo* with different *B. pertussis* isolates (Tohama, *B. pertussis* isolates FR5730, FR6440, FR4930, FR5333, FR5862; range $2.06 \times 10^7$ to $3.5 \times 10^8$ CFU/mL) and by *E. coli* (S88; $1.93 \times 10^6$ to $6.725 \times 10^7$ CFU/mL). Analytes were measured in CB plasma/supernatants after 22 hours of infection using a panel of 19 cytokines/chemokines and growth factors (Human Luminex Assay R&D Systems, Minneapolis). Cytokines are expressed as pg/mL and log transformed with blue to red colors representing lower to higher expression, respectively. (**A**) The heatmap represents a hierarchical clustering of all the cytokines/ chemokines and growth factors secreted in non-stimulated (NS) CB samples (in black) versus in CB samples stimulated by *E. coli* (in green), and by different *B. pertussis* isolates: Tohama (in pink), isolates from mild clinical forms (in red) and isolates from severe clinical forms (in dark red) (adjusted p-value = 1). (**B**) The heatmap represents a hierarchical clustering of all the cytokines/ chemokines and growth factors differentially secreted in CB samples stimulated by *E. coli* (in green; S88: $1.93 \times 10^6$ to $6.725 \times 10^7$ CFU/mL)) and by *B. pertussis* isolates (in red; Tohama, FR5730, FR6440, FR4930, FR5333, FR5862; $2.06 \times 10^7$ to $3.5 \times 10^8$ CFU/mL) with an adjusted p-value < 0.05. (**C**) Cell death following CB stimulation with *E. coli* with a range of bacterial inoculum (in CFU/mL; x-axis). Cell death rate (%) is represented in mean and standard deviation. Data representative of three experiments. (**D**) Cytokine release following *ex vivo* infection of CB samples (n = 6) with *E. coli* (strain S88, $1.93 \times 10^6$ to $6.725 \times 10^7$ CFU/mL). Analytes were measured in CB plasma/supernatants after 22 hours of infection using a 19-analyte panel (Human Luminex Assay R&D Systems, Minneapolis). Cytokines/chemokines are expressed as pg/mL and log transformed with blue to red colors representing lower to higher expression, respectively. The heatmap represents a hierarchical clustering of all cytokines/ chemokines and growth factors differentially secreted (adjusted p-value<0.05) in CB samples stimulated by *E. coli* (in green), compared to non-stimulated CB samples (NS, in black).
(DOCX)

**S3 Fig. Flow gating strategies and cell populations identification.** (**A**) List of antibody's surface-protein-targets and associated immune cell subsets used for spectral-cytometry analysis. (**B**) Representative flow plots and gating strategy to identify major immune cell subsets in CB. (**C**) Representative t-distributed stochastic neighbor embedding (t-SNE) plots from a spectral-immunophenotyping analysis of concatenated CB samples. Each dot represents an immune cell, and, for each plot, red-labeled cells correspond to the indicated immune cell compartment as determined by the gating strategy presented in (**B**).
(DOCX)

**S4 Fig. mMDSCs loss of detection after stimulation with *B. pertussis*.** Representative spectral-cytometry plots showing the detection of mMDSCs in CB based on their expression of CD14 and CD33 markers, across the different experimental conditions, and pre-gated either way on Live+ cells or Dead+ cells.
(DOCX)

**S5 Fig. Representative flow-plots of CD25+B cells across the different experimental conditions of infection.** (**A**) Representative spectral-cytometry plots showing the expression levels of CD25 and HLA-DR on B cells from AB (blue) or CB (red), across the different experimental conditions and at different time. (**B**) Representative spectral-cytometry plots showing the expression levels of CD25 and CD14 on B cells from whole-CB and PBMC from CB in non-stimulated (NS) condition or following stimulation with *B. pertussis* for 22 hours.
(DOCX)

**S6 Fig. Principal Component Analyses (PCA) of the gene expression profiles.** The plot shows the first three PCA axes based on all studied genes, with point colors representing gene groups. Each axis label includes the percentage of inertia, indicating the information captured by that axis. (**A**) Comparison of gene expression between cord blood (CB) and adult blood (AB) gene groups. (**B**) Comparison of gene expression between *B. pertussis* (FR4930), non-stimulated (NS) and stimulated gene groups.(**C**) Comparison across five independent experiments to assess batch effects. (DOCX)

**S1 Table. Characteristics of selected *B. pertussis* isolates.** (DOCX)

**S2 Table. Eligibility criteria for cord blood and adult blood donations.** (DOCX)

**S3 Table. Cell count for whole blood stimulation.** (DOCX)

**S4 Table. Luminex Panels.** (DOCX)

**S5 Table. Antibodies and dyes used for flow cytometry.** (DOCX)

**S6 Table. Expansion of the Nanostring Host Response panel with additional genes.** (DOCX)

**S1 Dataset. Excel files with extensive cytokine data.** (XLSX)

**S2 Dataset. Excel file with extensive transcriptomic data.** (XLSX)

## Acknowledgments

We are grateful to Nathalie Jolly, Mohand Ait-Ahmed (Centre for Translational Science) of Institut Pasteur, France for support in regulatory and ethical compliance. We thank Pr Stéphane Bonacorsi for providing the *E. coli* S88 strain.

## Data and materials availability

We generated all data set and provided extensive cytokine ("S1 dataset.xls") and transcriptomic datasets in our Supporting information files ("S2 dataset.xls"). The protocol of the study and ethical approvals are available on the laboratory website (research.pasteur.fr/en/project/fr-pert-severe-definir-la-contribution-des-facteurs-microbiens-et-de-limmunite-de-lhote-a-la-gravite-de-la-coqueluche/).

## Author contributions

**Conceptualization:** Julie Toubiana.

**Data curation:** Soraya Matczak, Mirko Sadi, Pierre Tonnerre, Julie Toubiana.

**Formal analysis:** Soraya Matczak, Mirko Sadi, Pauline Leroux, Pauline Labé, Sandrine Schmutz, Valentina Libri, Valérie Seffer, Sophie Novault, Milena Hasan, Darragh Duffy, Pierre Tonnerre, Julie Toubiana.

**Funding acquisition:** Julie Toubiana.

**Investigation:** Soraya Matczak, Mirko Sadi, Pauline Leroux, Pauline Labé, Valérie Bouchez, Sandrine Schmutz, Valentina Libri, Valérie Seffer, Sophie Novault, Sylvain Brisse.

**Methodology:** Stevenn Volant, Pierre Tonnerre, Julie Toubiana.

**Project administration:** Julie Toubiana.

**Resources:** Valérie Bouchez, Sylvain Brisse.

**Software:** Stevenn Volant.

**Supervision:** Mirko Sadi, Milena Hasan, Darragh Duffy, Pierre Tonnerre, Julie Toubiana.

**Validation:** Soraya Matczak, Mirko Sadi, Pauline Leroux, Pauline Labé, Valérie Bouchez, Sandrine Schmutz, Valentina Libri, Valérie Seffer, Sophie Novault, Stevenn Volant, Sylvain Brisse, Milena Hasan, Darragh Duffy, Pierre Tonnerre, Julie Toubiana.

**Visualization:** Soraya Matczak, Mirko Sadi, Pauline Leroux, Pauline Labé, Valérie Bouchez, Sandrine Schmutz, Valentina Libri, Valérie Seffer, Sophie Novault, Stevenn Volant, Sylvain Brisse, Milena Hasan, Darragh Duffy, Pierre Tonnerre, Julie Toubiana.

**Writing – original draft:** Soraya Matczak, Mirko Sadi, Pauline Leroux, Pauline Labé, Darragh Duffy, Pierre Tonnerre, Julie Toubiana.

**Writing – review & editing:** Soraya Matczak, Mirko Sadi, Pauline Leroux, Pauline Labé, Valérie Bouchez, Sandrine Schmutz, Valentina Libri, Valérie Seffer, Sophie Novault, Stevenn Volant, Sylvain Brisse, Milena Hasan, Darragh Duffy, Pierre Tonnerre, Julie Toubiana.

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
