## [Decision Letter · Decision Letter 0]

13 Oct 2025

PPATHOGENS-D-25-01905

Modeling neonatal immune response to B. pertussis identifies early B cell activation and differentiation

PLOS Pathogens

Dear Dr. Toubiana,

Thank you for submitting your manuscript to PLOS Pathogens. Your manuscript was evaluated by members of the editorial board and two experts whose comments are provided below. Both reviewers consider the work to have potential to make significant contribution to the fields of vaccinology against an important and re-emerging pathogen. Nevertheless numerous concerns were raised by both reviewers that require considerable revision of the manuscript to improve the clarity and rigor of the study, and should be carefully undertaken by you and your colleagues. Therefore, we invite you to submit a substantially revised version of the manuscript that addresses all of the points raised during the review process. In particular, there are a couple experiments requested by Reviewer 1 to enhance the impact of the findings, one relating to normalization of cell number and cell type, and another that would identify the cell type(s) responsible for the observed cytokine profiles.

Please submit your revised manuscript within 60 days Dec 12 2025 11:59PM. If you will need more time than this to complete your revisions, please reply to this message or contact the journal office at plospathogens@plos.org. Please include the following items when submitting your revised manuscript:

We look forward to receiving your revised manuscript.

Kind regards,

Victor J. DiRita

Guest Editor

PLOS Pathogens

D. Scott Samuels

Section Editor

PLOS Pathogens

Sumita Bhaduri-McIntosh

Editor-in-Chief

PLOS Pathogens

orcid.org/0000-0003-2946-9497

Michael Malim

Editor-in-Chief

PLOS Pathogens

orcid.org/0000-0002-7699-2064 

**Journal Requirements:**

At this stage, the following Authors/Authors require contributions: Soraya Matczak, Mirko Sadi, Pauline Leroux, Pauline Labé, Valérie Bouchez, Sandrine Schmutz, Valentina Libri, Valérie Seffer, Sophie Novault, Stevenn Volant, Sylvain Brisse, Milena Hasan, Darragh Duffy, Pierre Tonnerre, and Julie Toubiana. Please ensure that the full contributions of each author are acknowledged in the "Add/Edit/Remove Authors" section of our submission form.

- TM on pages: 16, and 17.

5) We have noticed that you have uploaded Supporting Information files, but you have not included a complete list of legends. Please add a full list of legends for your Supporting Information file(Supplementary appendix) after the references list.

Potential Copyright Issues:

i) Figure 1. Please confirm whether you drew the images / clip-art within the figure panels by hand. If you did not draw the images, please provide (a) a link to the source of the images or icons and their license / terms of use; or (b) written permission from the copyright holder to publish the images or icons under our CC BY 4.0 license. Alternatively, you may replace the images with open source alternatives. See these open source resources you may use to replace images / clip-art:

7) Since your data is not available for proprietary reasons, please explain via email why the data is not available. Please also include the contact information for the third party organization that should be contacted should other researchers want to request access to this data and please include the full citation of where the data can be found. We also request that you verify with us via email that any researcher will be able to obtain the data set in the same manner that the you have obtained it.

If you feel you are unwilling or unable to adhere to this policy, please explain your reasons by return email and your exemption request will be escalated to the editor for approval. Your exemption request will be handled independently and will not hold up the peer review process, but will need to be resolved should your manuscript be accepted for publication. One of the Editorial team will be in touch if they require more information.

8) Please amend your detailed Financial Disclosure statement. This is published with the article. It must therefore be completed in full sentences and contain the exact wording you wish to be published.

1) State what role the funders took in the study. If the funders had no role in your study, please state: "The funders had no role in study design, data collection and analysis, decision to publish, or preparation of the manuscript.".

**Reviewers' Comments:**

Reviewer's Responses to Questions

**Part I - Summary**

Reviewer #1: Bordetella pertussis infections are resurging despite global vaccination efforts. As the authors highlight, Bp infection is more severe in infants who have yet to receive any vaccination. In an effort to explore these differences, the authors develop an ex vivo infection model taking both adult blood and cord blood and treating them with different strains of Bp, highlighting differences between the immune responses in the different sources of blood. The authors highlight that while the cord blood is naturally suppressive in nature, that after Bp infection in their ex vivo model, there is a hyperinflammatory signature in cord blood. Similar in both blood sources was the activation of CD25+ B cell responses. Although this work could make a significant contribution to the field, there are several inconsistencies, unclear points, and aspects where the data lack sufficient context or relevance.

Reviewer #2: The manuscript “Modeling neonatal immune response to B. pertussis identifies early B cell activation and differentiation” by Matczak et al. utilizes an ex vivo whole-blood infection model of cord blood and adult blood to provide valuable new knowledge on age-specific immune responses to B. pertussis. This study is novel and generated information that will be of great interest in the field. The applied studies are robust, and the manuscript is clearly written. More information on experimental conditions is needed, and additional improvements are suggested.

**Part II – Major Issues: Key Experiments Required for Acceptance**

Please use this section to detail the key new experiments or modifications of existing experiments that should be absolutely required to validate study conclusions.required to validate study conclusions.

Reviewer #1: 1. It is noted that the authors measured the IgG responses to PT to determine whether the patient had recent exposure or an active infection. Vaccination history of the maternal parent should be included, as well as immunization history. Even though low titers describe an individual that is not immunized, literature shows that prior immunization can dictate the immune responses, specifically the T cell responses. If this information can not be obtained from the patients, this should be highlighted in the discussion, as this could potentially impact the results. Additionally, the impact of the study can be significantly improved if cohorts who are both aPV and wPV immunized could be examined or if maternal immunization strategies had any impact on the cord blood.

2. Further details of the strains are needed here. The information that is provided essentially highlights that they are the same. Additionally, countries that aPV are experiencing many PRN deficient strains.

3. More information is needed from the individuals that the blood was collected from. For example, any medication or drugs that the individual is on needs to be noted.

From the methods that are described, a volume of blood was used, not a cell count. I think it is necessary to have some sort of normalization of the cell number and types when presenting this data. This also makes the data reproducible when highlighting the amount of bacteria used. Furthermore, the manuscript as a whole would have a greater impact if the cell type responsible for the cytokine profiles that are observed were determined. What cells are specifically contributing to the cytokine profiles. Intracellular flow cytometry on the specific innate and adaptive immune cells would determine the relative contribution of each cell type.

Lines 72-73. Although Fig. 1 lists multiple B. pertussis strains, in the cytokine/chemokine analysis the authors focus on FR4930 without explaining why this strain was selected for the initial analyses. The authors should clarify.

Fig. 2 (cytokine/chemokine analysis). The control (non-stimulated) samples display high baseline levels of several cytokines/chemokines (e.g., sICAM ~10⁴ pg/ml, IL-12p70, IL-12, IFN-γ). Moreover, IFN-γ levels appear indistinguishable between stimulated and non-stimulated conditions, yet the authors describe it as “specific to stimulation” in line 78. The authors should explain why the non-stimulated samples show such high baseline levels and discuss the biological relevance of these findings. There also appears to be a discrepancy between the differences observed in non-stimulated versus stimulated samples in Fig. 2A and the levels shown in Fig. 2B.

Basal cytokine/chemokine levels are not comparable between CB and AB samples in fig 2A, but the authors do not indicate which basal levels were used to generate Fig 2B.

Fig 2, line 109. The term “expression” is not appropriate in this context. Cytokine/chemokine measurements represent secreted levels, not gene or protein expression. The wording should be corrected accordingly.

Fig 2, line 110. The authors mention that samples were organized by increasing bacterial concentrations and by groups along the y-axis, but the grouping strategy and actual concentrations are not clearly indicated in the figure or in the Results section. This omission makes interpretation difficult. The authors should clarify how groups were defined and ensure that concentrations are explicitly shown in the figure. In addition, it is not clear which concentration was used to generate Fig 2A.

Supplementary Fig S1. It is not indicated how cell death was evaluated, either in the figure or in the Materials and Methods. This information is essential for interpreting the results. Cytokine/chemokine-producing immune cells circulating in blood, such as monocytes and lymphocytes, can survive for more than 24 h ex vivo, whereas other populations such as neutrophils could have a much shorter lifespan (~6–17 h). Despite this, no reduction in viability is reported after 22 h post-infection. The authors should clarify how cell death was assessed and explain these observations.

Line 90. The authors should clarify the rationale for including an E. coli strain in the Results section, as this would facilitate interpretation of the comparative findings.

Line 96. The authors state that B. pertussis induced a stronger inflammatory response compared to E. coli; however, this difference may be confounded by the higher levels of cell death observed with E. coli.

Reviewer #2: (No Response)

**Part III – Minor Issues: Editorial and Data Presentation Modifications**

Reviewer #1: (No Response)

Reviewer #2: 1. The methods do not clearly specify the volumes of blood used per well in each experiment, nor whether cell concentrations (e.g., cells per mL) were normalized across samples, citing only that 200 to 500 ul of whole blood samples were used. This is particularly important given that the study compares whole cord blood (CB) and adult blood (AB), which can differ in cell numbers. Indeed, the finding that cytokine responses in CB were more elevated at a protein level than what was observed for transcriptional responses may be due to increased cell numbers in the CB resulting in higher amounts of secreted proteins, while transcript measurements are normalized to the expression of housekeeping genes. The authors should provide additional information on the volumes of blood used per experiment and whether cell concentrations were normalized. If these factors were not controlled, a discussion of the potential impacts of differences in cell numbers per well should be added.

2. The assay used to measure cell death is not described in the methods, please add this information.

3. A discussion of the potential impact of comparing different infectious doses for B. pertussis and E. coli would improve reader comprehension.

4. The described figure annotations for Supplementary Fig S2 are incorrect (line 95-97). S2C displays cell death measures, not S2B; S2D displays E. coli-induced inflammation not S2C; S2B compares B. pertussis with E. coli, not S2D.

5. Information on the source (CB or AB or both) of control (NS) blood displayed in Fig 2B needs to be added.

6. INFg should be changed to IFNg on line 116, 287 and Fig 1.

7. Both moMDSC and mMDSC are used e.g. line 120 and line 125. This should be standardized.

8. The numbers of samples per group should be given for Fig 4B, as there appear to be overlapping samples or just 1 sample was evaluated for % CD25+ B cells in AB stimulated with Tohama. If this group is n=1, statistical analysis should not be applied.

9. The labels for Fig 6B, 6C and 6D are incorrectly assigned on the figure as compared with the description in the legend, this should be corrected.

10. The loss of CD16 is described as a marker of granulocyte activation here, however loss of surface CD16 expression is associated with degranulation or apoptosis of neutrophils, with increased CD16 associated with activation (e.g. see PMID:23228566 and PMID:10380913). In addition, with the currently provided information, it is not clear whether this shift in expression is due to a cell population shift or phenotype shift; CD66b+ CD16- cells may reflect altered neutrophil states or an increased proportion of eosinophils which are CD16-. Adding clearer labels to the flow gating strategy displayed in Fig S3B and a complete list of marker composition for each cell in S3A (e.g. Treg: CD45+ CD3+ CD19- CD25+ CD127+) will provide necessary details for the reader to understand the possible cell populations included in each group. While additional discussion on the different mechanisms of reduced CD16 surface expression would also improve the manuscript.

11. The methods used for determining pathway enrichment should be provided.

12. The heatmap headers for Fig 5A and 6A are not consistent or easily understood, I suggest revising these.

PLOS authors have the option to publish the peer review history of their article (what does this mean?). If published, this will include your full peer review and any attached files.). If published, this will include your full peer review and any attached files.

.

Reviewer #1: No

Reviewer #2: No

**Figure resubmission:**
---

## [Editor Report · Decision Letter 1]

3 Mar 2026

PPATHOGENS-D-25-01905R1

Modeling neonatal immune response to B. pertussis identifies early B cell activation and differentiation

PLOS Pathogens

Dear Dr. Toubiana,

Thank you for submitting your manuscript to PLOS Pathogens. After careful consideration, we feel that it has merit but does not fully meet PLOS Pathogens's publication criteria as it currently stands. Therefore, we invite you to submit a revised version of the manuscript that addresses the points raised during the review process.

* A letter that responds to the minor point raised by the editor below. You should upload this letter as a separate file labeled 'Response to Reviewers'. This file does not need to include responses to any formatting updates and technical items listed in the 'Journal Requirements' section below.

We look forward to receiving your revised manuscript.

Kind regards,

Victor J. DiRita

Guest Editor

PLOS Pathogens

D. Scott Samuels

Section Editor

PLOS Pathogens

Sumita Bhaduri-McIntosh

Editor-in-Chief

PLOS Pathogens

orcid.org/0000-0003-2946-9497

Michael Malim

Editor-in-Chief

PLOS Pathogens

orcid.org/0000-0002-7699-2064

**Additional Editor Comments:**

Thank you for providing a substantially revised manuscript in response to concerns that were raised by the reviewers. You have effectively addressed the major points raised. As we discussed, there was a concern about one of your citations and you indicated that this would need to be corrected through a minor revision. Please submit a suitably modified version with that section re-written accordingly. Specifically, you were going to modify as shown below.

“At baseline, ICAM levels were elevated, as previously described (Duffy et al., Immunity. 2014, PMID: 24656047). When cytokine concentrations from stimulated CB and AB were directly compared, 10 showed differential secretion (Fig 2B); of these, five were significantly upregulated upon stimulation: VEGF-A was found more elevated in AB whereas levels of IL-1a, IL-1b, IL-12p40 and TNF-a were higher in CB after stimulation with B. pertussis. IFNγ levels were higher in CB than in AB, but the stimulation-specific effect did not reach statistical significance, likely due to elevated baseline IFNγ levels and inter-donor variability (Biancotto et al, Plos one, PMID: 24348989; Duffy et al., Immunity. 2014, PMID: 24656047)."

**Journal Requirements:**

1) In the online submission form, you indicated that Other raw data from our study are available upon reasonable request. All PLOS journals now require all data underlying the findings described in their manuscript to be freely available to other researchers, either

1. In a public repository

2. Within the manuscript itself

3. Uploaded as supplementary information.

2)  Please ensure that the funders and grant numbers match between the Financial Disclosure field and the Funding Information tab in your submission form. Note that the funders must be provided in the same order in both places as well.

3) Please ensure that the figure legend includes the proper citation indicating the figure was created using BioRender.

**Reviewers' Comments:**

**Figure resubmission:**While revising your submission, we strongly recommend that you use PLOS’s NAAS tool (https://ngplosjournals.pagemajik.ai/artanalysis) to test your figure files. NAAS can convert your figure files to the TIFF file type and meet basic requirements (such as print size, resolution), or provide you with a report on issues that do not meet our requirements and that NAAS cannot fix.
---

## [Editor Report · Decision Letter 2]

9 Apr 2026

Dear Prof. Toubiana,

We are pleased to inform you that your manuscript 'Modeling neonatal immune response to B. pertussis identifies early B cell activation and differentiation' has been provisionally accepted for publication in PLOS Pathogens.

Best regards,

Victor J. DiRita

Guest Editor

PLOS Pathogens

D. Scott Samuels

Section Editor

PLOS Pathogens

Sumita Bhaduri-McIntosh

Editor-in-Chief

PLOS Pathogens

orcid.org/0000-0003-2946-9497

Michael Malim

Editor-in-Chief

PLOS Pathogens

orcid.org/0000-0002-7699-2064

Thank you for making the minor edits and revisions we requested. I think this work will be an excellent citation for you and for the journal, and appreciate your sending it to PLOS Pathogens.
---

## [Editor Report · Acceptance letter]

Dear Prof. Toubiana,

We are delighted to inform you that your manuscript, "Modeling neonatal immune response to B. pertussis identifies early B cell activation and differentiation," has been formally accepted for publication in PLOS Pathogens.

Best regards,

Sumita Bhaduri-McIntosh

Editor-in-Chief

PLOS Pathogens

orcid.org/0000-0003-2946-9497

Michael Malim

Editor-in-Chief

PLOS Pathogens

orcid.org/0000-0002-7699-2064